# Design of 3D Scaffolds for Hard Tissue Engineering: From Apatites to Silicon Mesoporous Materials

**DOI:** 10.3390/pharmaceutics13111981

**Published:** 2021-11-22

**Authors:** Ana García, María Victoria Cabañas, Juan Peña, Sandra Sánchez-Salcedo

**Affiliations:** 1Departamento de Química en Ciencias Farmacéuticas, Facultad de Farmacia, Universidad Complutense de Madrid, UCM, Instituto de Investigación Hospital 12 de Octubre, i+12, 28040 Madrid, Spain; anagfontecha@ucm.es (A.G.); vcabanas@ucm.es (M.V.C.); juanpena@ucm.es (J.P.); 2Networking Research Center on Bioengineering, Biomaterials and Nanomedicine (CIBER-BBN) Madrid, 28040 Madrid, Spain

**Keywords:** rapid prototyping, GELPOR3D, silica-hydroxyapatite, mesoporous materials, meso–macroporous scaffolds, ceramic modifying agents

## Abstract

Advanced bioceramics for bone regeneration constitutes one of the pivotal interests in the multidisciplinary and far-sighted scientific trajectory of Prof. Vallet Regí. The different pathologies that affect osseous tissue substitution are considered to be one of the most important challenges from the health, social and economic point of view. 3D scaffolds based on bioceramics that mimic the composition, environment, microstructure and pore architecture of hard tissues is a consolidated response to such concerns. This review describes not only the different types of materials utilized: from apatite-type to silicon mesoporous materials, but also the fabrication techniques employed to design and adequate microstructure, a hierarchical porosity (from nano to macro scale), a cell-friendly surface; the inclusion of different type of biomolecules, drugs or cells within these scaffolds and the influence on their successful performance is thoughtfully reviewed.

## 1. Introduction

As is well known, bone tissue engineering requires the use of tridimensional (3D) scaffolds which act as temporary templates for cells to guide bone repair, stimulating natural mechanisms of bone regeneration [1]. These processes are affected by both hierarchical pore structure and the chemical surface composition, which constitute fundamental pillars in the design and fabrication of these scaffolds. However, most synthetic 3D scaffolds lack chemical surface properties involved in the biological recognition processes and in the early stage of cell adhesion, necessary to ensure the complete cell colonization for subsequent bone regeneration [2,3].

Already in the 2008 editorial article entitled “Current trends on porous inorganic materials for biomedical applications” that introduced the Special Issue devoted to “Porous Inorganic Materials for Biomedical Applications” [4], Prof. Vallet-Regí expressed the necessity “… to induce in these materials porosity in the range of microns so that they can fulfill physiological requirements in their use as scaffolds for tissue engineering”. This sentence reflected her already widespread experience with clinical scientists regarding the in vivo implantation of the materials prepared by her group.

At that moment, the term “macropore” became the IUPAC category that comprised those pores bigger than 50 nm to those in the hundreds of microns range that, in addition, should be interconnected to induce vascularization and tissue ingrowth. Moreover, the word hierarchical was already mentioned expressing the necessity of a pore distribution in the mesoporous and even in the microporous range together with the bigger pores already discussed.

Besides the pore architecture concerns, another critical point in scaffold fabrication has traditionally been the preparation of hybrid materials by combining different components as ceramic and hydrated organic substances that resemble the composition and microstructural features of bone, one of the tissues that demand these type of supports to regenerate different types of osseous defects [5]. Taking into account that mechanical support constitutes one of the main functions of the musculoskeletal system, the requirement of an appropriate balance between mechanical properties and porosity is a key parameter in the design of such substitutes.

Finally, the importance of working at room temperature was outlined, i.e., preparing the scaffolds under physiological conditions that allow direct inclusion of labile biomolecules or following a biomimetic approach, synergistically combining the different components that should resemble the natural tissue to be regenerated. However, the frequent incompatibility of many fabrication methods with these mild conditions has induced the creation of several strategies implying scaffold modification in a second step in order to include drugs, natural biomolecules and, even, cells.

In general, for any type of tissue, a scaffold should have the following characteristics [6]:-It should possess a 3D interconnected porous hierarchical architecture.-It should possess a suitable surface chemistry and topography for cell attachment, proliferation and differentiation.-It should be tailored to include substances whose controlled release may contribute to the integration of the scaffolds without any adverse reactions.-It should be biocompatible and bioresorbable with controllable degradation and resorption rates to match tissue replacement without any undesirable by-products.-It should present mechanical properties that match those of the tissue during the reconstruction process. In addition, these scaffolds should be consistent enough to allow their manipulation during the cell seeding or surgical implantation procedures and even to be adapted in situ to fit odd-shaped defects.-It should show full reproducibility under large-scale manufacturing conditions.-Additionally, it should have a long shelf life and/or be straightforwardly preservable and be easily available to surgeons in a sterile operating environment.

This article attempts to compile the considerable effort carried out by Prof. Vallet-Regí’s group to answer to all these concerns by employing diverse and smart approaches related to the fabrication technologies, the porosity generation and the scaffolds upgrading and tailoring to improve their performance.

This research was carried out by using, mainly, two designed controlled scaffold fabrication methods (Figure 1): rapid prototyping [7,8,9,10,11,12,13,14,15,16,17,18,19,20,21,22] and GELPOR3D or analogous procedures [23,24,25,26,27,28,29,30,31,32,33,34,35]. In addition, foams have been synthesized by combining the sol–gel technique with the addition of a surfactant as a random macropore former that is thermally eliminated; the so-obtained inorganic foams are coated with gelatin or polycaprolactone (PCL) yielding tougher and more biocompatible scaffolds [36,37,38,39,40]. Besides this preparation route, additional non-designed porosity generation methods: freeze drying and porogen leaching/calcination have been employed to create supplementary porosity within the scaffolds fabricated by the above mentioned pore designable technologies.

For the manufacture of 3D scaffolds with designed porosity, the rapid prototyping technique has been extensively employed during the last three decades [41,42,43,44]. This technique is based on direct ink writing, where the filament obtained is extruded from a small nozzle connected to a cartridge as it is moved across a platform so that regular scaffolds are built, by printing the required shape, layer by layer. The rapid prototyping technique reproduces a previously computer-aided design format by injecting a paste with the right rheological properties and using a robot injector (Figure 1). In order to obtain ceramic inks with properties suitable for extrusion, it is necessary to add several substances, many of them polymeric, which confer adequate parameters of fluidity and viscosity. Depending on the suspension composition the obtained scaffolds require an ulterior drying/consolidation step which is usually followed by the elimination of the polymeric additives by means of a thermal treatment or its dissolution. Alternatively, the removal of such components is eluded when the objective is preparing hybrid scaffolds. In these cases, the substances added to facilitate the ceramic paste extrusion must be, in addition, biocompatible and/or biodegradable [45,46].

The GELPOR3D method consists of the solidification of a suspension containing a thermogel and an inorganic component. Consolidation of the pieces is achieved inside a mold by means of a thermogelling agent, such as agarose or gellan, which is able to gel at physiological temperature. In order to obtain interconnected macroporous pieces, solidification is carried out using a network of filaments designed to ensure interconnected porosity in the three spatial directions. The network of filaments directing this structure can be easily extracted at room temperature without aggressive solvents or any other type of subsequent treatment (Figure 1). This fabrication technology allows preparation of scaffolds containing different ceramics, polysaccharides and one, or more, products such as therapeutic agents, biomolecules, etc. Any of the molecules included in situ show a much-sustained release when compared to those incorporated by impregnation in a second step [31,32]. Indeed, the inclusion of nanoparticles within these scaffolds not only prolongs the dosage effect but also allows control of it as a function of a particular pH, temperature, light or magnetic variation/stimulus. Concerning the ceramic stability, although the absence of a thermal treatment ensures the maintenance of the microstructural and textural properties, those highly soluble/reactive inorganic fillers may be modified during their suspension in an aqueous solution; however, neither the bioactive glasses nor the mesoporous ceramics are affected in their microstructure and properties [23,29].

The “second-step” or ex situ inclusion of a substance within an already fabricated piece is a well-known and widespread technology that allows improvement of the scaffolds’ final performance by, for example, including one or more drugs with different release patterns or transforming the surface into a cell-friendly environment. As shown in Figure 1 this additional stage is essential for scaffolds prepared by any technique that requires an aggressive chemical or thermal treatment to eliminate certain substances or to consolidate the obtained piece. This has been the strategy followed to incorporate different biomolecules: rifampicin, levofloxacin, zoledronic acid, osteostatin, etc., within scaffolds prepared by rapid prototyping [15,16,17,19,20]. In a similar way, the scaffolds’ surface has been modified through its functionalization with different types of biomolecules. Moreover, coating with gelatin or collagen constitutes a direct and easy way to change the surface characteristics and also to load certain drugs/biomolecules; this strategy has been also applied to improve scaffolds obtained by GELPOR3D [28] and also in the case of foams (Figure 1) [38,39,40].

Moreover, it must be stressed that alternative approaches that avoid the critical thermal or aggressive chemical step have enriched the applicability of prototyping and similar technologies. For example, organic polymers such as PCL or polyvinylalcohol (PVA), that are employed to facilitate the generation of a printable paste, are maintained yielding hybrid organic/inorganic scaffolds. The presence of these, FDA-approved, biocompatible substances has uncovered new possibilities to load biomolecules and/or to functionalize the surface.

Concerning the generation of the “ideal” hierarchical pore architecture reflected in Figure 2, different simultaneous or sequential procedures have been employed. The bigger/giant pores (ultra-macro scale), in the hundreds of microns range, can be previously designed according to a certain pattern which ensures its interconnection in the three dimensions and are obtained through the rapid prototyping extrusion or by means of the removable filaments in the GELPOR3D. On the other extreme, micro or mesopores are given by the inorganic component included; nevertheless, the different fabrication steps must be designed and applied to avoid any alteration of the microstructural and textural properties of these ceramics. The interval between these two pore populations, from 50 nm to 100 µm, which is critical for the permeation processes, nutrient supply, waste removal, can be generated by different methods such as lyophilization or porogen extraction/dissolution. Freeze-drying or lyophilization is a preservation technology (frequently employed in the food and pharmaceutical industry) that generates great porosity due the extraction of the solvent crystals. The freeze-drying of hydrogels like those employed in the fabrication of scaffolds (agarose, gellan…) generate parallel pores of around 100 µm that resemble a honeycomb. This microstructure is maintained even in the presence of the high amount of the ceramic particles aggregated by these polysaccharides. Porogen extraction, traditionally through a thermal treatment, can be carried out at room temperature by dissolving the particles included in the scaffold matrix [22] or a polymeric network used as a sacrificial mold [23,24].

This paper is a review article in honor of the scientific career of Prof. Vallet-Regí on scaffolding for the bone regeneration area in the last 20 years. This review tries to condense the part of her prolific scientific trajectory devoted to the design of tridimensional scaffolds that could efficiently solve the current pathologies which arise for clinicians. This manuscript has been organized in terms of the function of inorganic matrices, calcium phosphates, glasses and mesoporous glasses, which constitutes the main component of designed, fabricated, in vitro and in vivo evaluated implanted scaffolds. Figure 3 summarizes the different ceramic scaffolds described in this review and compiles the different modifications and/or incorporations conducted in these ceramic matrices.

## 2. Calcium Phosphate Scaffolds

Bone is the major calcification present in the human body. From the chemical point of view, bone is a composite or hybrid material. Calcium phosphates are the most important inorganic constituents of biological hard tissues (bones and teeth) of mammals. Due to their chemical similarity with biological calcified tissues, all calcium phosphates generally show excellent biocompatibility; they are accepted well by the body. Consequently, they found important applications as biomaterials, particularly for hard-tissue regeneration [48,49,50,51].

Calcium phosphates are a group of bioceramics and, from the point of view of biomaterials for bone tissue, the most important are hydroxyapatite (HA), tricalcium phosphate (β-TCP) and biphasic calcium phosphate (BCP), i.e., a composite of HA and β-TCP. In reality, the mineral component of the bones does not correspond to a stoichiometric HA, Ca_10_(PO_4_)_6_(OH)_2_. Biological apatites in mammals are calcium-deficient and nonstoichiometric, and contain carbonate ions, CO_3_^2−^, in their crystalline structure which replace PO_4_^3−^ groups. They are known as carbonated hydroxyapatites (CHA). This substitution contributes to the higher solubility of biological apatites compared to stoichiometric HA. The presence of carbonate ions helps to keep the constant bone regeneration through dissolution–crystallization processes.

These calcium phosphates have been synthesized and characterized by the Prof. Vallet-Regí’s group both in the powder form and as coating by using different synthetic methods [52,53,54,55,56,57,58]. Afterwards, the challenge of the group has been to obtain porous ceramics that act as scaffolds for cells and signals molecules and able to drive self-regeneration of bone tissue. Although other bioceramic scaffolds such as β-TCP [24,27] and BCP [25,26,35,59] have been also prepared by this research group, HA has a pivotal role in the scaffold preparation.

As we will discuss in this section, the research group has used different shaping methods, which allow tailoring the porosity and shape of the scaffolds, as well as to design different chemical strategies to increase their functionality (Figure 4). The purpose has been to adapt these ceramics fabricated as porous scaffolds for using in tissue engineering as well as to achieve the surgeon’s requirements.

The rapid prototyping technique, Figure 1, has been used to fabricate three-dimensional HA scaffolds, with a computer designer shape [60]. These HA scaffolds were manufactured from a slurry prepared from stoichiometric HA, synthesized in our laboratory [53,61], and the PCL polymer, which acted as a binder during the solid free forming. After that, the scaffolds were calcined to remove the PCL in order to obtain pure HA 3D scaffolds (Figure 5a,b). The total porosity of the fabricated scaffolds was 64%, of which 48% corresponded to pores smaller than 10 μm, 4% to pores in the 10–100 μm range and 12% to pores in the range between 100 and 600 μm. The cellular response to a biomaterial in terms of both adhesion and proliferation on its surface is a fundamental requisite in order to be used as a support in tissue engineering. These HA scaffolds showed high biocompatibility in vitro as demonstrated by cell culture with HOS osteoblastic cells. The study showed viable and adequate osteoblastic-like cell spreading which have preserved their typical osteoblast morphology, exhibiting polygonal shapes with small filopodia like projections anchored to the surfaces (Figure 5c) [60].

In order to fabricate pure HA scaffolds in the form of spheres, Vila et al. used a sol–gel technique in combination with a hydrogel as a template [62]. Following the hydrolysis of a solution containing Ca and P, in a Ca/P ratio = 1.67, sodium acrylate–acrylamide copolymer beads were immersed in this sol resulting in an increase in the sphere’s volume to its maximum. The removal of the organic template by thermal treatment generates a pure HA scaffold with a porosity ranging between 1 and 500 μm. The HA spheres showed an adequate in vitro degradation performance and an enhanced in vitro bioactivity compared to synthetic HA.

However, considering the composition of bones, HA combined with other polymeric components, such as collagen, gelatin, diverse polysaccharide, etc., would be the ideal material to use in bone tissue engineering [63,64,65,66,67]. Among the polymers, both natural and synthetic hydrogels present many similarities with the macroscopic macromolecular components of the extracellular matrix of living beings [68,69]. In this sense, as described below, the research group has fabricated numerous HA composite-based scaffolds.

Hybrid scaffolds composed of apatite/polysaccharide were fabricated by Peña et al. by the GELPOR3D technique [29] (Figure 1). The polymeric component, the polysaccharides agarose or gellan, serves as a binder for the consolidation of the scaffold and imparts a hydrogel nature, which resembles the environment and composition of the extracellular matrix. The ceramic component, CHA, was synthesized in our laboratory showing some features similar to biological apatites [70]. The CHA/polysaccharide scaffold manufacturing process [30] consists of heating an aqueous agarose or gellan suspension (2.5–5% *wt/v*) to around 90 °C; subsequently, the working temperature is decreased to 40 °C and the ceramic CHA (20–80%) is added to the suspension. This slurry is injected into the designed mold and, after complete consolidation, a 3D scaffold is obtained (Figure 5d). These scaffolds can be easily manipulated and shaped with a blade or scalpel which allows the clinician to adapt the size and shape of the piece to the osseous defect. These pieces behave like hydrogels with water content around 90–95% and can be preserved by two drying procedures: in an oven at 37 °C and by freeze-drying. After the drying process, the scaffolds maintain their shape but suffer a slight shrinkage which is higher when the drying is carried out in an oven. Their hydrogel behavior determines that it can absorb a fluid (plasma or blood), swell without destruction, and maintain its overall structure [71]. This feature allows the material to fit perfectly to the defect exerting a constant pressure on the walls of the same. This small pressure can act as a mechanical stimulus for osteoblast activity. As has been mentioned, porosity is an important property of bone, as it allows the body fluids and cells to access the various regions of the osseous. Román et al. [30] demonstrated that these CHA/polysaccharide scaffolds can be hierarchically organized: the macroporosity of these scaffolds is determined, fundamentally, by the drying procedure; lyophylized scaffolds are constituted by designed ultra-macro pores (between 700 and 900 μm as a function of the ceramic content) together with 100–200 μm-sized pores organized within a honeycomb or sponge-like structure (Figure 5e). The oven-dried samples are constituted by giant pores between 600 and 800 μm within a wrinkled collapsed structure since the water elimination does not lead to the formation of the honeycomb structure. The CHA content also contributes to modify these porosities; the lowest porosities are observed when the ceramic content increases.

The in vitro biocompatibility of these CHA/polysaccharide hybrid scaffolds was demonstrated by culturing Saos-2 osteoblastic cells onto the surface of these biomaterials [28,31]. The cells adhered, proliferated and colonized the macroporous scaffolds showing a high cell viability. These scaffolds allow osteoblast growth while suffering neither plasma membrane damage nor oxidative stress. The scaffold surface shows a high hydrophilicity which favors the adhesion and proliferation of preosteoblastic cells. In some cases, osteoblasts on the scaffold presented a spherical shape which suggested a loss of cell/biomaterial adhesion, which was related to particles released from the biomaterial. The cell adhesion was improved by coating the scaffolds with a type I collagen. After this treatment, osteoblasts were spread well on the biomaterial surface and presented the appropriate morphology (Figure 5f). Furthermore, the CHA/agarose scaffolds showed a satisfactory biocompatibility when implanted subcutaneously in the Wister female rat model [33]. The macropores of the scaffold facilitated cell intrusion and the production of extracellular matrix within the scaffold. Moreover, the progressive degradation of the material contributes to the formation of new bone tissue. The CHA/agarose scaffold is capable of promoting in vivo osteogenesis, thus improving connective integration.

Another strategy used by the Prof. Vallet-Regí’s group in the manufacture of inorganic/organic-based scaffolds was the fabrication of biopolymer-coated HA foams. Sánchez-Salcedo et al. [37] manufactured 3D HA foams (Figure 5g) by the sol–gel technique including a non-ionic surfactant, Pluronic F127, as a macropore inducer. After sol aging the mixture is treated at 100 °C, to evaporate the solvent, and posteriorly is calcined at 550 °C to remove the surfactant. In order to confer flexibility and handleability the 3D-HA foams were coated with different biocompatible polymers approved by FDA: Gelatine crosslinking with glutaraldehyde or PCL. These HA foams showed a high volume of interconnected macroporosity in the 1–400 μm (Figure 5h). The in vitro degradability tests carried out on these scaffolds exhibited a gradual dissolution of the foam over time, but maintain their 3D shape and integrity even after 30 days of incubation. The dissolution rate can be controlled and slowed down when the foam is coated with a biopolymer, gelatin/glutaraldehyde [36]. The 3D interconnected architectural design of these HA foams allows excellent osteoblast internalization, proliferation and differentiation, exhibiting adequate colonization over the entire scaffold surface with an appropriate degradation rate without any cytotoxic effects (Figure 5i). The potential of these gelatin-coated HA scaffolds as bone regenerators in vivo was studied in mature male New Zealand rabbits [38]. In this study, a critical bone defect was made in the lateral aspect of distal femoral epiphyses, which was filled with the gelatin-coated HA foams. Their swelling ability allows them to perfectly fill the bone defect cavity when hydrated, leading to good fixation without the need for cement. The HA scaffolds showed osteointegration, bone growth over its surface and subsequent remodeling after 4 months of implantation. They were progressively infiltrated by blood from the surrounding healthy bone, allowing cell colonization. The interconnected porosity of the scaffold allowed 3D colonization of the defect, avoiding calcification of the material.

In addition to the aforementioned requirements of scaffolds, regarding chemical properties and pore architecture (Figure 2), the recreation of a favorable microenvironment to increase the formation of bone requires the release of biologically active molecules at the target site for a given period of time. As discussed below, Prof. Vallet-Regí’s group has also taken up this challenge by introducing different active substances for tissue restoration in the different types of HA scaffolds fabricated (Figure 3). The delivery of therapeutic substances or biomolecules such as growth factors, active peptides or cytokines to guide cellular growth is considered an effective strategy to design biomaterials for regenerative medicine [72,73]. In this sense, the significance of bone morphogenetic proteins to increase the osteoinductive properties of implanted scaffolds is well recognized [74]. The C-terminal 107–111 fragment of parathyroid hormone-related protein (PTHrP), known as osteostatin, presents osteogenic features in vitro and stimulates bone accrual in vivo [75,76]. Taking this into account, Lozano et al. [40] loaded gelatin-coated HA foams with synthetic osteostatin, by immersing them in a saline solution of this peptide. After 24 h of soaking, the mean uptake of osteostatin by these scaffolds was around a 60% which is readily released in a short period of time. These loaded scaffolds were implanted into a cavitary defect made in distal tibial metaphysis of adult rats. The obtained results demonstrated that the presence of osteostatin onto HA scaffolds increases bone healing with profuse osteoblastic cells adhering to the trabecular surface in the proximity of the degrading biomaterial and reduced the profusion of osteoclasts resorbing new bone around the implant. Moreover, these coated HA foams have also been employed as a template for PTHrP (1–37) or PTHrP (107–111) delivery and to evaluate the efficiency of such peptides to promote bone regeneration in a scenario combining aging and Diabetes Mellitus, frequently related to bone fragility and increased fracture risk [39]. This work demonstrated that the local delivery of either PTHrP peptide from degradable HA implanted into a rat’s noncritical tibial defect neutralizes the adverse effects imposed by age and diabetes on bone regeneration.

The introduction of these active biomolecules into the scaffold presents several drawbacks such as high cost, denaturation or degradation of protein structure during scaffold fabrication and adverse side effects when overdosed. Consequently, a lot of effort has been exerted to incorporate biological molecules into the scaffolds. As has been shown in the previous paragraph, in general, they are introduced into the scaffold by adsorption. However, as was previously mentioned, the GELPORD3D technique allows the inclusion of these active substances into the scaffold during the preparation process as well as to control the amount introduced. Thermally labile molecules can be incorporated into CHA/polysaccharide systems affecting neither the gelation process nor the capacity to easily shape the scaffolds [31,32]. Cabañas et al. included Bovine Serum Albumine (BSA), used as model protein, into CHA hybrid scaffolds by two procedures [31]: (i) during the fabrication process, a solution containing the protein was added to the slurry (in situ process); (ii) BSA was included after scaffold preparation, taking advantage of the hydrogel character of the scaffolds (ex situ process). It must be emphasized that in both processes a precise control of the BSA amount incorporated into the scaffold is achieved and, due to the mild fabrication conditions, no denaturalization of the protein was produced. Both procedures showed different protein release kinetics: the BSA in situ inclusion showed much lower releasing rates when compared to the equivalent ex situ samples.

Moreover, a fundamental step in bone regeneration is the blood vessel formation [77,78]. The inclusion of pro-angiogenic factors such as Vascular Endothelial Growth Factor (VEGF) can induce the successful colonization of these porous scaffolds by blood vessels to ensure nutrient for the cells growing on it [79,80]. In order to promote the vascularization of the engineered scaffold, Paris et al. [32] added a precise amount of VEGF into CHA/agarose scaffolds during their fabrication process. The biological activity of VEGF released from scaffold was studied using an ex ovo chicken embryo’s chorioallantoic membrane model. The study performed demonstrated an increase in blood vessels, especially small capillaries, around the CHA/agarose scaffold.

On the other hand, the simultaneous release within the target site of two or more therapeutic substances (dual drug delivery systems) with different pharmacological activity, should improve the outcome in different bone pathologies [81,82]. For example, zoledronic acid, a bisphosphonate anti-osteoporotic drug, is clinically applied for the treatment of diseases related to increased bone resorption [83,84,85]. Ibuprofen, an anti-inflammatory drug, is commonly prescribed concurrently with bisphosphonates to minimize one of its main side-effects [84,86]. Both therapeutic substances were successfully incorporated in a controlled manner into CHA/agarose scaffolds and co-delivered to the medium following diverse tunable release profiles [34]. The zoledronic acid presented a sustained release, essential to promoting bone regeneration, whereas a burst delivery of ibuprofen (3 h), too fast for the clinical application, was observed. In order to extend ibuprofen release, it was encapsulated in chitosan particles before being embedded into the scaffold. After this modification, the burst effect is minimized and its release lasts more than two days [34].

A significant hurdle for tissue regeneration often arises from bacterial infection, which mitigates the tissue healing/repair process and results in limited regeneration [87,88,89,90]. Thus, it is motivating to develop bone regeneration scaffolds that can simultaneously prevent the onset of infection and/or treat it where it occurs. For this purpose, different strategies have been designed by using HA-based scaffolds, such as the doping with metal ions with antibacterial properties (e.g., Ag^+^, Cu^2+^, Zn^2+^, …) [91,92,93,94] or the design of hybrid scaffolds with intrinsic antimicrobial activity (e.g., chitosan-based scaffolds [95,96,97]). Alternatively, this challenge has also been addressed by Prof. Vallet-Regí’s group by using two different approaches which allow one to combat bone infection and simultaneously to regenerate bone tissue (Figure 6): (i) by using surface modification technologies that impart antiadherent properties for prevention, (ii) by incorporating enough antibiotic within the scaffold to reach a local release for treatment of bacterial infection. In the first case, Sánchez-Salcedo et al. used HA scaffolds, fabricated by rapid prototyping, to build very hydrophilic surfaces able to inhibit bacteria attachment [60]. For this purpose, pure HA scaffolds were post-synthesis bifunctionalized with -NH_3_^+^ and -COO^−^ groups to provide them with a zwitterionic surface (Figure 6A). *Escherichia coli* was used as a bacteria model to evaluate the in vitro bacterial adhesion. This research demonstrates the versatility of the functionalization procedure in preparing 3D zwitterionic HA. The efficient interaction of these zwitterionic pairs on the HA surface reduces the *E. coli* adhesion more than 90% and at the same time the in vitro biocompatibility of these zwitterionic HA scaffolds was preserved.

In the second approach, scaffolds suitable for bone regeneration and loaded with antibiotics, the research group designed CHA/agarose scaffolds fabricated by GELPORD3D considering that this method allows an easy incorporation of the antibiotic [32] (Figure 6B). A precise amount of cephalexin was included in in this hybrid scaffold during the fabrication process. The release profiles of cephalexin from these hybrid scaffolds, in a physiological solution, corresponds to a first-order kinetic. The agar diffusion test indicated that the cephalexin did not lose its activity during the scaffold fabrication and scaffolds delivered an antibiotic local concentration capable of inhibiting *Staphylococcus aureus* bacterial growth (Figure 6B(I)).

Although most of the results above-described indicate that the inclusion of therapeutic substances within these scaffolds has shown promising results, the complications in achieving a proper control over drug release stands as an unresolved issue and new strategies have been developed. In this sense, nanoparticles have been proposed to control the release of different substances. This approach has been used in our group in the development of scaffolds for bone regeneration that can simultaneously prevent bone infection, combining two major research topics of the Prof. Vallet-Regí group. The research group has extensively studied Mesoporous Silica Nanoparticles (MSNs) which are biocompatible, nontoxic and excellent drug delivery vehicles with controllable releases [98,99,100,101,102]. Paris et al. [32] used these MSNs to load cephalexin and these encapsulated drug-MSNs were included into the slurry used for the preparation of CHA/agarose scaffolds. The inclusion of these MSNs within scaffolds is a powerful tool that allows us to achieve a sustained release of the cephalexin and thus to prolong the antibiotic treatment (Figure 6B(II)). Moreover, the silicon ions present in these MSNs can be released to the surrounding media, hence stimulating osteogenesis and angiogenesis [103], both important issues in the development of biomaterials for the successful regeneration of bone tissue as discussed in the following section.

## 3. Silicon-Substituted Hydroxyapatite (Si-HA) Scaffolds

Hydroxyapatite is well-known to be bioactive and osteoconductive material and, as described in the previous section, it has been extensively investigated. In order to increase a patient’s quality of life it is essential to achieve rapid bone tissue repair and this process can be enhanced by improving the bioactive properties of HA. One strategy has been the silicon incorporation in HA formulation. In this sense, Si has been included in the HA lattice by replacing some phosphate groups by silicate groups resulting in silicate substituted HA formulated as Ca_10_(PO_4_)_6−x_(SiO_4_)_x_(OH)_2−x_. Silicon is an essential trace element required for healthy bone and connective tissues and influences the biological activity of calcium phosphate materials. Since the 1970s, the role of Si in the development of skeletal tissue and the formation of new bone has been established mainly due to the fact that silicon is associated with calcium in an early stage of calcification [104]. Carlisle described that a relatively high concentration of Si is present in metabolically active osteoblasts and is considered to be essential in the formation of extracellular matrix in bone and cartilage [105]. In addition, a Si reduction in bone results in a decrease in the number of osteoblasts [106], the amount of osseomatriceal collagen, and glycosaminoglycans [107]. It has been proven that the presence of Si affects the cellular response at the implant/bone interface, which influences the rate of bone tissue repair. Over the last two decades, Prof. Vallet-Regí has successfully contributed to the development of silicon-doped hydroxyapatite (Si-HA) material [108,109,110,111,112,113,114] and the conformation of these bioceramic matrices in three-dimensional supports that promote bone tissue growth. For the manufacture of Si-HA in the form of 3D scaffolds, her research has focused on the technique of rapid prototyping.

Prior to manufacturing of 3D scaffolds, it was necessary to synthesize Si-HA as a powder. Prof. Vallet-Regí research group has prepared Si-HA with the nominal formula Ca_10_(PO_4_)_6−x_(SiO_4_)_x_(OH)_2−x_□ where x = 0.25, 0.3 and 0.5 and □ indicates vacancies at the hydroxyl position. Si-HA powders were prepared by aqueous precipitation reaction of Ca(NO_3_)_2_·4H_2_O, (NH_4_)_2_HPO_4_ and Si(CH_3_CH_2_O)_4_ solutions, as described elsewhere [111].

In order to prepare the slurry with the adequate properties to be extruded, the methodology used by the Prof. Vallet-Regí research group consisted of mixing the bioceramic powder with organic polymers that act as binders. Once the 3D scaffolds were manufactured by the rapid prototyping technique (Figure 1), two procedures were followed: to remove the organic phase thus obtaining pure ceramic scaffolds or to maintain the organic component and take advantage of polymer properties to improve the final hybrid system. Table 1 summarizes the main research strategies developed by Prof. Vallet-Regí’s group to prepare 3D scaffolds of a Si-HA ceramic matrix by the rapid prototyping method and the principal results obtained.

In order to obtain pure Si-HA, scaffolds the inks were prepared by using a mixture of monomers: methacrylamide (MMA), N,N′-methylene-bis-acrylamide (MBAA), ammonium persulphate (acting as the initiator) and surfactant Darvan^®^ 811 [7,9,115,116], or by mixing Darvan^®^, hydroxypropyl methylcellulose and polyethylenimine as the flocculant [117]. The organic phase employed during the processing stage was removed by calcination, thus obtaining pure ceramic scaffolds (Figure 7). Matesanz et al. [117] described pure Si-HA scaffolds obtained through 700 °C and 1250 °C thermal treatment. Scaffolds treated at high temperature improved mechanical properties in comparison to those treated at 700 °C. The low cell response observed in Si-HA scaffolds treated at 700 °C versus 1250 °C could be due to the topography different of the surface microstructure. The scaffolds treated at 700 °C resulted in Si-HA materials with higher surface areas and porosities that facilitated protein adsorption (for example, albumin and fibrinogen). These scaffolds exhibited an adequate interaction with pre-osteoblast and osteoblast cells. The treatment at 1250 °C Si-HA resulted in highly crystalline biphasic α-TCP/HA (15/85%) materials which improved pre-osteoblast-like cell proliferation and differentiation, but a low adsorption of proteins onto these scaffolds was observed, which induced a loss of cell anchorage over poorly sintered surfaces.

Commonly, biomaterial implants include not only the porous scaffolds but also the growth factors, functional drugs, peptides, proteins, biologically active molecules or osteogenic cells that could stimulate and even accelerate the bone repair process [122,123]. Baeza et al. [7] functionalized the surface of Si-HA scaffolds with biotin molecules by covalent bonding through the esterification reaction between polyhydroxyl groups present in the scaffold surface due to HA presence and the carboxylic groups of biotin (Figure 3). Si-HA biotinylation opened the possibility of anchoring a great number of proteins or peptides or drugs by a simple chemical procedure.

Decorated pure Si-HA scaffolds with biologically active molecules were used development by Prof. Vallet-Regí’s research group as successful strategies for bone tissue engineering. These molecules are able to interact with the surrounding bone tissue, hence improving biological response in terms of osseointegration and scaffold resorption. Manzano et al. [115] described the preparation of osteostatin-decorated Si-HA scaffolds through two working methodologies: (i) scaffolds with adsorbed osteostatin where the electrostatic interactions at physiological pH were responsible for the osteostatin coating of this material, (ii) Si-HA scaffolds with covalently bound osteostatin where the covalent grafting of osteostatin was performed through the formation of an amide bond between the hydroxyl groups of its C-terminal and the amino groups previously incorporated on the Si-HA scaffold surface. The exposure of pre-osteoblastic MC3T3-E1 cell monolayers to Si-HA scaffolds decorated with osteostatin, both deliverable and covalently immobilized, stimulated cell growth and matrix mineralization, which is expected to promote bone formation in vivo.

Cell growth factors are considered among the most attractive types of bioactive agents for bone tissue engineering due to their important roles in skeletal development and postnatal osteogenesis. Fibroblast growth factors (FGFs) are a group of polypeptides whose expressions have been associated with bone development, growth and repair [124,125]. FGF-1 and FGF-2 are acid and basic fibroblast growth factors, respectively, and both are considered to be representative of the whole FGF family. In addition, neovascularization plays a main role in bone development and consequently in the design and engineering of implantable tissues because a rapid induction of angiogenesis is required to meet the significant oxygen and nutrient demands of cells during tissue repair processes. FGFs also act as strong inducers of angiogenesis [126]. Feito et al. [9] incorporated FGF-1 or FGF-2 to Si-HA scaffolds through non-covalent binding by soaking the scaffolds with each growth factor solution. Simple adsorption of either FGF-1 or FGF-2 on the bioceramic scaffold led to an improved adhesion and proliferation of Saos-2 osteoblastic cells onto Si-HA scaffolds, confirming the potential utility of these FGF/Si-HA scaffolds for bone tissue engineering.

Combining Si-HA and the Elastin-like Recombinamers (ELRs) polymers was another strategy to enhance the biological response of this ceramic material. ELRs are based on repeated sequences, namely ‘‘elastomeric domains”, found in an extracellular matrix protein, the natural elastin [127,128]. Vila et al. [116] worked with ELR-RGD, which contained cell attachment specific sequences, that induces migration and adhesion osteoblast thus improving the bone regeneration processes [129,130] and ELR-SN_A_15/RGD, that enclosed both HA and cell domains interacting with the inorganic phase and with the cells, respectively. The SN_A_15 domain of statherin, whose interaction with calcium phosphate is well-established [131], has a fragment of 15 amino acids located at the N-terminus that exhibits a higher affinity for Si-HA compared to that shown by the whole statherin molecule due to the negative charge density and the helical conformation which exists in this domain [132,133,134]. The in vitro tests performed on rapid prototyping Si-HA scaffolds biofunctionalized by ELRs adsorption showed that a complete and homogeneous colonization of the scaffolds by bone marrow mesenchymal stromal cells took place. The presence of the integrin-mediate adhesion domain (RGD) alone or in combination with the SN_A_15 peptide did not determine an increase in cell proliferation rate and, in both cases, similar viability was observed. Instead, the ability of inducing mesenchymal stem cells differentiation into the osteoblastic lineage was statistically different. Differentiation studies reveal a positive effect of SN_A_15/RGD to guide the mesenchymal stem cells into the osteoblastic lineage.

PVA was also used as a binder to prepare the adequate slurry to conform scaffolds by rapid prototyping. Casarrubios et al. [118] removed this organic polymer at 1150 °C. As a consequence of this high temperature treatment, the microstructure of Si-HA was modified and pure nanocrystalline ceramic scaffolds (nanoSi-HA) were obtained. These scaffolds exhibited high surface area and porosity. Furthermore, with the aim to improve their biological activity, their surfaces were decorated with VEGF by means of a simple impregnation method and tested in vitro in osteoporotic sheep. The nanoSi-HA scaffolds showed a poor response in vivo, in terms of new osseous regeneration after 12 weeks of implantation. Nanocrystalline microstructure led to poor bone ingrowth; thin trabeculae reduced presence of osteoblasts and produced a high presence of osteoclasts. The association of VEGF with nanoSi-HA scaffolds displayed better results in terms of the volume of newly formed bone, trabeculae thickness and implant vascularization.

As mentioned above, another strategy followed by Prof. Vallet-Regí’s research group consisted of maintaining the organic part used for the inks’ preparation remained together with the Si-HA. In these cases, it is a necessary requirement that the organic polymers used to facilitate the scaffold manufacturing are biocompatible and/or biodegradable. The Prof. Vallet-Regí research group has used three polymers for this strategy: gelatine, PCL and PVA.

Martínez-Vázquez et al. [119] reported the fabrication of Si-HA/gelatine scaffolds by rapid prototyping in a single step where the type A-gelatine (porcine skin) was dissolved in water and mixed with Si-HA powder at room temperature. Due to the presence of gelatine, these scaffolds behaved like a hydrogel. The easiness of manipulation and shaping, thanks to its hydrogel nature, constitutes a great advantage for its handling during the surgery and for its adaptation to the bone defect. In addition, the mechanical properties of such scaffolds were similar to those of trabecular bone of the same density while, at the same time, in vitro studies with pre-osteoblastic MC3T3-E1 cells revealed several benefits of gelatine inclusion in the ceramic material in terms of cell proliferation and differentiation (matrix mineralization and gene expression). On the other hand, the presence of gelatine facilitated the incorporation of therapeutic agents (in this case vancomycin) in the slurry before the extrusion of the scaffolds. Vancomycin released to the medium gradually inhibited bacterial growth around the material. In this way, 3D scaffolds with dual activity were obtained: a support with good properties for bone regeneration thanks to macro and micropore architecture but also a system for drug(s) loading and release.

PCL is a biocompatible polymer with potential applications for bone and cartilage repair approved by FDA polymer [135,136,137]. This polymer has long degradation time due to its high degree of crystallinity and hydrophobicity and exhibits non-toxicity effects in vivo [138]. In fact, scaffolds exclusively based on PCL have been proposed as implants for bone tissue repair [139]. However, some authors have reported on the convenience of incorporating osteogenic agents (BMP-7, BMP-2, etc.) [140,141] or HA [142,143] to overcome the poor osteoconductive behavior of PCL.

Meseguer-Olmo et al. [120] described the preparation of Si-HA/PCL scaffolds in which their walls were formed by a fibrous continuous phase that corresponds to organic polymer, together with a discrete phase that corresponds to the Si-HA ceramic component (Figure 7). Moreover, in order to improve this system, a demineralized bone matrix (DBM) that is considered as a potential osteoinductive material [144,145] was included. When Si-HA/PCL scaffolds were implanted in the tibiae of New Zealand rabbits, worthy results in terms of biocompatibility, osteoconductive features, and new bone formation capability could be observed. However, when the DBM was incorporated, the efficacy of the Si-HA/PCL/DBM scaffolds was enhanced, as new bone formation occurred not only in the peripheral portions of the scaffolds but also within their pores after 4 months of implantation. This superior performance could be explained in terms of the osteoinductive properties of the DBM in the scaffolds, which have been assessed through the new bone tissue formation when the scaffolds were ectopically implanted.

Scaffolds constituted by PCL, different weight percentages of Si-HA (from 50 to 40 wt.%) and carbon nanotubes (CNTs) (from 0 to 10 wt. %) were described by Gonçalves et al. [121]. After the Si-HA/PCL/CNTs scaffolds were manufactured by rapid prototyping technique, electron microscopy characterization revealed that the Si-HA and CNTs phases were homogeneously dispersed in the PCL polymeric phase, suggesting that the organic polymer was embedded the inorganic phases in the composite (Figure 7). When the inorganic particles were examined in detail, CNTs and Si-HA maintained their tubular and acicular morphology, respectively. In order to evaluate the effect of the CNT content on the mechanical properties of the Si-HA/PCL/CNTs scaffolds, the stress of the yield at the first inflexion point was analyzed to determine the compressive yield strength, or the load-bearing capacity of the composite materials [146]. Si-HA/PCL/CNTs scaffolds with 0.75% CNTs displayed an increase in the compressive yield stress to 6.5 MPa relatively to the unreinforced Si-HA/PCL material (4.2 MPa). Considering that the composites become electrically conductive with 2% CNTs, this scaffold offered the best combination of mechanical behavior and electrical conductivity. This composite presented, on one side, a compressive strength of ~4 MPa, similar to that of trabecular bone, but also allows the application of electric stimuli for bone healing purposes. Concerning in vitro response it was observed that, for all compositions, MG63 osteoblast-like cells were attached at the scaffolds’ surface, this fact being more significant in pieces with 10% of CNTs, which was in agreement with the fact that CNTs’ presence improved protein adsorption and, consequently, cell attachment.

PVA is a biocompatible and biodegradable polymer, which has been used in different bio-applications as drug release, tissue engineering and enzyme immobilization [147,148,149]. Casarrubios et al. [118] described Si-HA/PVA scaffolds that were heated at 150 °C to induce heat-activated PVA cross-linking, without modifying the microstructure of the inorganic component. With the aim of improving its biological activity, VEGF was incorporated to the surface by impregnation. Non-covalent interaction took place because VEGF has a strong affinity by these kind of bioceramics [150] and Casarrubios et al. demonstrated that after 96 h this molecule was still retained almost completely. In vitro cell culture tests showed that the microstructure and functionalization with VEGF stimulated endothelial (EC_2_) proliferation and pre-osteoblasts (MC3T3-E1) differentiation, thus resulting in enhanced vascularization and new bone formation observed in vivo as deduced from the formation of thicker trabeculae and higher presence of osteoblasts after twelve weeks osteoporotic sheep implantation.

Continuing with the scenario of bone tissue regeneration, Prof. Vallet-Regí research group proposed using Si-HA as a coating for macroporous metal scaffolds (Ti6Al4V-ELI) [151]. Metal scaffolds facilitate osteogenesis and new blood vessels formation within their macroporous structure, while exhibiting optimal mechanical behavior, which are two critical aspects for the regeneration of bone defects. Ti6Al4V-ELI scaffolds with 3D interconnected macroporosity were prepared by electron beam melting and were coated with Si-HA by means of a dip-coating process, where different synthesis parameters (mainly ageing times) were strictly controlled to prepare homogeneous and stable coatings. Izquierdo-Barba et al. suggested that the presence of immobilized VEGF together with Si-HA would result in a better angiogenic response and stimulation of new bone formation from both the inner site and peripheral area of the bone defect. In order to test this proposal, the in vitro response of endothelial (EC_2_) and pre-osteoblastic (MC3T3-E1) cells to the above-mentioned Si-HA@Ti6Al4V-ELI scaffolds was evaluated. In vitro cell culture tests evidenced that scaffolds with adsorbed VEGF stimulated proliferation of endothelial cells on the scaffolds’ surface, whereas scaffolds coated only with Si-HA stimulated proliferation of preosteoblasts. In order to evaluate the osteogenic capability of the Si-HA@Ti6Al4V-ELI scaffolds, with or without VEGF, in conditions simulating osteoporosis in humans, Izquierdo-Barba et al. implantation in an osteoporotic sheep model was carried out. After 12 weeks of implantation, clear differences were observed between different scaffolds. In vivo studies proved that neither Si-HA coating nor VEGF adsorption enhance osteogenesis separately (new bone formation restricted to peripheral regions was observed). However, the VEGF adsorption on Si-HA-coated scaffolds exhibited a synergistic effect resulting in increased ossification, larger trabeculae formation and a higher angiogenesis degree with a further developed vascular system. VEGF reinforced the osteoinductive capability of Si-HA-coated scaffolds, which were able to stimulate new bone formation even in the inner sites of the defect not in contact with the peripheral bone.

## 4. Bioactive Glass (BG) Scaffolds

Other very interesting materials for bone tissue regeneration, which Prof. Vallet-Regí has researched extensively over the last 20 years, are the so-called bioactive glasses (BGs) [152,153]. BGs formed by melt or sol–gel techniques are considered highly reactive surfaces forming an apatite-like layer when immersed in physiological fluids (or solutions that simulate human plasma) which enhance protein adsorption to the surface of the implant and establish a rapid, strong, and stable bond with host tissues, promoting new bone-tissue formation at their surface when implanted in the living body [154,155,156,157]. Prof. Vallet-Regí’s group focused their research on the BG scaffolds with a molar composition of SiO_2_(55%)-CaO(45%) and SiO_2_(55%)-CaO(41%)-P_2_O_5_(4%) prepared by the sol–gel method, where precursors of the corresponding oxides were tetraethyl orthosilicate, calcium nitrate tetrahydrate and triethyl phosphate, respectively. In order to shape these BGs as 3D pieces with the designed architecture, the research group developed different methodologies: combining the gelcasting method and the stereolithography technique and the GELPOR3D method (Figure 1).

Cabañas et al. [23] used a biodegradable polysaccharide (agarose) as a binder agent dissolved in water and SiO_2_(55%)-CaO(45%) as a bioactive ceramic. They designed porous pieces by using an epoxy polymer negative template, previously designed by stereolitography (Figure 1). Scaffolds containing around 85 wt.% of inorganic component and 15 wt.% agarose were obtained by pouring the aqueous slurry into polystyrene templates. The negative polymeric mold was subsequently eliminated by alkaline dissolution (NaOH 2 M) at room temperature for 24 h obtain scaffolds with an interconnected network of channels (350 µm) and macropores (0.5 µm).

Padilla et al. [158] described a 3D scaffold manufacturing by casting a homogeneous suspension of BGs into the templates which are polymerized afterwards. On this occasion, the slurry was prepared by mixing the dispersant vehicle (Darvan^®^ 811) and the SiO_2_(55%)-CaO(41%)-P_2_O_5_(4%) powder with a solution of monomers composed of a 15 wt.% MMA and MBAA, ammonium persulfate (as initiator) and *N*,*N*,*N*′,*N*′-tetramethylendiamine (as a catalyst). The slurry was casted into the negative templates and polymerized (by the gel casting method). After the pieces dried, the organic templates were removed by heat treatment. The polymeric template was removed at 1100 °C and the scaffolds were sintered at 1300 °C, obtaining pieces with interconnected porosity, 3D channels of 400–470 µm and macropores around 1–2 µm.

A few years later, Peña et al. [29] described an alternative method, GELPOR3D, to shape scaffolds with hierarchical porosity under simple conditions using a low-cost equipment, compositional flexibility, and a lack of aggressive or toxic solvents or other high thermal treatment. BG scaffolds generated by the GELPOR3D technique were prepared from SiO_2_(55%)-CaO(45%) powder and an aqueous solution of agarose as a binder.

Despite the methodology employed to fabricate the scaffolds, the bioactivity role of the ceramic matrices was hardly affected and a new apatite-like layer was developed over the scaffolds’ surfaces. It is true that, when the NaOH solution was used to remove the negative polymeric template [23], small alterations were detected by the presence of crystalline CaSiO_3_ and Na_2_CO_3_ phases compared to scaffolds obtained from thermal treatment [158] or the GELPOR3D technique [29]. The in vitro bioactivity response of the obtained pieces has driven the choice of the BGs and later mesoporous bioactive glasses (MBGs) as ceramic materials for the purpose of reaching complete substitution by natural bone and evaluating the satisfactory restoration of the original biological performance.

## 5. Mesoporus Bioactive Glass (MBG) Scaffolds

The inspiration for the synthesis of MBGs were sol–gel glasses, but also pure silica mesoporous materials designed for the first time in the 1990s in the effort to design materials able to improve the performance of zeolites in catalysis [159]. Searching for a porous structure and high surface area of mesoporous materials (with very low bioactive response) and the high bioactivity of sol–gel glasses, a new class of intermediate materials between both of them was synthesized. Compared to the previously described bioceramics, MBGs present an ordered mesoporous arrangement, with high surface areas and pore volumes that allow their utilization as drug control release systems for bone tissue disease treatment. Furthermore, the high amount of free silanol groups on their surface presents a promising strategy for the anchoring of several substances through a covalent attachment, thus enhancing the applications of these materials.

In 2001, Prof. Vallet-Regí proposed the application of silica mesoporous materials as matrixes in drug delivery systems [160]. Chen et al. prepared highly ordered MBGs with superior in vitro bone-forming bioactivities for the first time in 2004 [161]. Since then, thousands of articles investigating this property have been published. In this context, Prof. Vallet-Regí’s group have made a considerable effort and widely research MBG scaffolds as optimum candidates for bone regeneration. In this sense, different 3D scaffolding methods have been optimized allowing achievement of hierarchically MBG porous 3D-printed scaffolds (Figure 2). Table 2 summarizes the main research strategies developed by her group to prepare 3D MBG scaffolds by the rapid prototyping method and the main results obtained.

The first time that Prof. Vallet-Regí’s group published a new method to prepare meso–macroporous 3D scaffolds was by employing the GELPOR3D technique in 2010 [29]. The mesoporous phase was SBA-15 with an ordered mesoporous structure with a pore size of 8 nm and high specific surface area of 817 m^2^/g [162]. Freeze-dried porous pieces containing 55 wt.% SBA-15 and 45 wt.% agarose were prepared by the procedure described above (Figure 1). The preparation allowed the creation scaffolds shaped with a hierarchical porosity which ranged from macro or ‘‘giant” interconnected pores bigger than 600 µm to the mesopores characteristic of this SBA-15 ordered structure.

In 2011, García et al. [8] developed a new system that consists of the preparation of 3D mesoporous glass scaffolds in the SiO_2_–P_2_O_5_ system with hierarchical meso-macroporosity by using a one-step sol–gel method and rapid prototyping technique. The synthetic method consists of the combination of a single-step sol–gel route in the presence of a surfactant (Pluronic F127) as the mesostructure directing agent and a biomacromolecular polymer such as hidroxy methylcellulose as the macrostructure template followed by the rapid prototyping technique. After the calcination, a hierarchical meso–macroporous system was obtained. With the same scaffold fabrication method, Cicuéndez et al. [11] developed 3D scaffolds based on a new nanocomposite based on the SiO_2_–P_2_O_5_–CaO system with HA nanocrystals embedded into mesoporous glass (MGHA) obtained after the calcination process. They exhibit three scales of porosity meso (10 nm), large-macro (1–80 µm), and ultra-large macropores (500 µm). The surface properties of these MGHA scaffolds, concerning cell recognition, have been enhanced by means of a direct functionalization with aminopropyl groups. This simple and cost-effective amine modification induces remarkable improvements in preosteoblast adhesion, a two-fold increase in cell proliferation and a four-fold increase in cell differentiation. No alterations in proliferation and viability of RAW-264.7 macrophage-like cells were observed after being cultured on MGHA scaffolds which did not generate cell apoptosis.

In 2012, Prof. Vallet-Regí’s group patented the preparation of 3D meso–macroporous scaffolds from MBGs by extruding pastes with PCL and hydroxyl methylcellulose using a rapid prototyping technique and then drying and calcining between 600 and 1000 °C for 4 h [163] (Figure 1). Following this patent strategy but avoiding the calcination step, Gómez-Cerezo et al. [14] surveyed new insights on preosteoblast behavior in MBG/PCL scaffolds prepared by rapid prototyping, by assembling individual strati composed by 58SiO_2_-36CaO-6P_2_O_5_ (% mol) MBGs. The in vitro cell colonization is fortified by the presence of this MBG ternary system, as confirmed by the evaluation of the cell behavior within the different scaffolds levels, i.e., from the initial source of cells towards the further scaffold locations. This MBG improves the cell migration towards upper strati through the dissolution of chemical signals and the changes that take place on the supports surface during the bioactive process. Furthermore, the MBG promotes preosteblastic proliferation and differentiation when compared to those scaffolds made of pure PCL. Finally, this study reveals the role of the architectural design to enhance cell colonization. These experiments shed light on the parameters that should be taken into account to accelerate the regeneration processes under in vivo conditions. For the first time, these meso–macroporous MBG/PCL scaffolds were evaluated in a sheep model that mimicked the osteoporosis conditions in humans [17]. These implants fostered bone regeneration, promoting new bone formation at both the peripheral and the inner parts of the scaffolds, showing thick trabeculae and a high vascularization degree (Figure 8). Moreover, in order to evaluate the effects of the local release of an antiresorptive drug in bone defects, zoledronic acid was added to the scaffolds. This drug inhibits macrophage differentiation into osteoclasts but also induced apoptosis in osteoblast-like cells. Under in vivo conditions, the incorporation of zoledronic acid inhibited the new bone formation and promoted a strong inflammatory response.

Subsequently, Gómez-Cerezo et al. [22] enhanced this MBG/PCL system by using a mixture of PCL, MBG and phosphate buffered saline (PBS) particles. 3D multiscale porous scaffolds with macro to mesoporosity were fabricated by using the rapid prototyping method. The resultant scaffolds displayed a highly interconnected macroporosity inherent to the additive manufacturing technique used, along with a microporous network with a 30% porosity while the mesoporosity was imparted by the presence of the MBG particles. The concomitant effects of the leaching of the PBS and the release of calcium from the MBG particles resulted in the formation of a crystalline calcium phosphate layer over the scaffold that significantly increased surface hardness. It was demonstrated that the modification in the biomechanical properties was driven by the presence of microporosity, as opposed to the incorporation of inorganic fillers that is conventionally seen in non-microporous 3D-printed scaffolds. The additional porosity increased cell metabolic activity due to the increase in surface area available for cellular interaction with the microporous scaffold and was further enhanced by the presence of the MBG particles.

Other systems designed by Gómez-Cerezo et al. [18] were centered on fabricating MBG/PVA macroporous fully interconnected scaffolds, using PVA as binder to facilitate the printing process by rapid prototyping. This water-soluble synthetic polymer is not only biocompatible, biodegradable, and nontoxic, but also provides sufficient mechanical strength and flexibility. However, these additives covered the MBG particles resulting in the reduction in their osteogenic potential. A simple and effective immersion in a PBS solution allows removal of the PVA binder while preserving the mesoporous structure of MBG 3D printed scaffolds. The extensive surface remodeling induced by the deposition of the apatite-like layer led to a three-fold increase in surface area, a five-fold increase in the roughness, and a four-fold increase in the hardness of the scaffolds immersed in the PBS solution when compared to the as-printed counterpart. The biomimetic mineralization also occurred throughout the bulk of the scaffold connecting the MBGs particles and was responsible for the maintenance of structural integrity. In vitro assays using pre-osteoblast like cells demonstrated a significant upregulation of osteogenic-related genes for the scaffolds.

Post-operative implant infection is one of the most serious concerns associated with surgical treatments of bone diseases and fractures by means of bone grafts and prostheses [60,164]. After their association, bacteria normally secrete polymeric substances that form protective coatings known as biofilms. Biofilms have been described as ‘‘aggregates of microorganisms in which cells are frequently embedded in a self-produced matrix of extracellular polymeric substances that are adherent to each other and/or a surface” [165]. These biofilms further hinder the activity of the host defenses and/or antibiotic therapy, involving surgical intervention to remove the implant as the only efficient alternative. In order to combat this situation, scaffolds have been designed to combat infection in bone defects.

Following this strategy, multifunctional-therapeutic MGHA nanocomposite 3D scaffolds were designed to demolish the *S. aureus* bacterial biofilm while allowing bone regeneration at the same time [16]. This research focused on the design of pH sensitive 3D hierarchical meso–macroporous 3D scaffolds based on a MGHA nanocomposite formed by HA nanocrystals embedded within a mesostructured glassy network whose mesopores had been loaded with levofloxacin as antibacterial agent. These 3D platforms exhibited a controlled and pH-dependent levofloxacin release, sustained over time at physiological pH (7.4) that notably increased when the pH is modified (6.7 and 5.5) as a consequence of an infection, which is due to a different interaction between levofloxacin and the silica matrix. These 3D systems were able to inhibit the *S. aureus* growth and to extinguish the bacterial biofilm without cytotoxic effects on human osteoblasts hence facilitating an adequate colonization and differentiation of preosteoblastic cells on their surface.

In the same context, García-Alvarez et al. [15] designed a hierarchical 3D multidrug scaffolds based on a bioceramic and PVA composite prepared by rapid prototyping with an external coating of gelatin-glutaraldehyde. These 3D scaffolds contained three antimicrobial agents (rifampicin, levofloxacin and vancomycin), which were localized in different compartments of the scaffold to obtain different release kinetics and more effective combined therapy. Levofloxacin was loaded into the mesopores of the composite bioceramic matrix, vancomycin was confined into the PVA biopolymer part and rifampicin was localized in the gelatin–glutaraldehyde external coating. These 3D multidrug scaffolds, showing an immediate and fast release of rifampicin followed by a sustained and prolonged release of vancomycin and levofloxacin, were able to demolish *Staphylococcus* and *Escherichia* biofilms as well as to inhibit bacteria growth in scarce time periods.

Likewise, with the aims of (i) improving bone osteogenic implants and (ii) fighting against microbial resistance and providing needed options for bone infection treatment, Prof. Vallet-Regí’s group enhanced these systems by enriching the MBG structures with inorganic therapeutic ions, and biomolecules (osteoinductors or antibiotics) inducing synergistic effects (Figure 3). On one side, these therapeutic ions that exhibit osteogenic, angiogenic and antimicrobial properties were added from the corresponding nitrate sources during the sol preparation. The combination of this doped MBG powder obtained by evaporation-induced self-assembly and the rapid prototyping technique were employed to fabricate 3D scaffolds using the binders described in previous sections. Prof. Vallet-Regí’s group mainly focused on the effect of the inclusion of Ga, Ce, Sr and Zn ions in MBGs [166]. Nevertheless, after the inclusion of extra elements in the CaO–P_2_O_5_–SiO_2_ system, it was necessary to confirm that the significant features that make MBGs useful for tissue engineering and drug delivery systems were maintained.

The 3D scaffolds based on MBGs in the system (85−x)SiO_2_–5P_2_O_5_–10CaO–xSrO (x = 0, 2.5 and 5 mol.%) with meso-macroporosity were fabricated by pouring a suspension of MBG powders in PVA into a negative template of polylactic acid (PLA), followed by the removal of the template by extraction at low temperature [21]. SrO-containing MBGs exhibited excellent properties for bone substitution including ordered mesoporous structure, high textural properties, quick in vitro bioactive response in simulated body fluid (SBF) and the ability to release concentrations of strontium ions able to stimulate expression of early markers of osteoblastic differentiation. Likewise, the direct contact of MC3T3-E1 pre-osteoblastic cells with the scaffolds confirmed the cytocompatibility of the three compositions considered. Furthermore, the investigated scaffolds exhibited cytocompatibility with pre-osteoblastic cells, especially those based on 2.5% SrO MBGs, in which the cytocompatibility was enhanced by the inductive capacity of strontium ions on proliferation and ALP gene expression of the cells. Consequently, it could be concluded that 2.5% SrO-MBGs/PVA scaffolds were an outstanding candidate to be fully developed to be employed for bone defect treatment in non-load bearing sites.

On the other hand, 3D MBG scaffolds based on 80%SiO_2_-15%CaO-5%P_2_O_5_ (in mol.%) mesoporous sol–gel glasses substituted with Ce_2_O_3_, Ga_2_O_3_ (both 0.2% or 1%) and ZnO (0.4% or 2%), were obtained by evaporation-induced self-assembly as powder and subsequently shaped by rapid prototyping [10]. All of them contained well interconnected ultralarge pores (>400 µm) and macropores of 100–400 µm were present inside the scaffolds. A slight decrease in surface area and pore volume was observed upon substitution with no significant variation on pore diameter. In addition, all the MBG scaffolds except that with a 2% ZnO content showed relatively quick in vitro bioactive response. With the aim of going further in the capacity to increase osteogenesis, angiogenesis, and antimicrobial properties of zinc ions, ZnO-enriched MBG scaffolds with 4 and 7% of ZnO were shaped by rapid prototyping [12]. MBGs scaffolds with 4% ZnO exhibited good biocompatibility behavior in terms of adhesion, proliferation, differentiation and the absence of cytotoxic degradation products. Moreover, 4 and 7% ZnO MBG scaffolds, under in vitro conditions simulating severe infection, exhibited 2.8- and 3.6-fold higher *S. aureus* inhibition capacity than non-zinc substituted MBG scaffolds. Thus, the composition and 3D-architecture of MBG allowed a controlled zinc release, which greatly influenced the osteoblast cell development. Furthermore, the ions released provided a bacterial growth inhibition capacity that would lead to a decrease in infection rates after implantation surgery. The multi-functional characteristics of the MBG scaffolds studied have been suggested to be of great potential to regenerate lost bone tissue overall with 4% ZnO MBG scaffolds. When these scaffolds [47] were loaded with osteostatin and decorated with human mesenchymal stem cells (MSCs), they exhibited a synergistic effect to enhance MSC growth and MSCs osteogenic differentiation without addition of other osteoblastic differentiation factors to the culture medium. This novel strategy has a great potential in bone tissue engineering. Moreover, the osteogenic capability of ZnO-enriched MBG scaffolds loaded with osteostatin and MSC was evaluated in vivo after implantation in New Zealand rabbits (Figure 8) [20]. Cylindrical meso–macroporous scaffolds with a 7 mm diameter and 10 mm depth were implanted in bone defects in the distal femoral epiphysis to evaluate and compare their osteogenic features. The systems investigated exhibited bone regeneration capability after 3 months. Prof. Vallet-Regí’s group demonstrated, for the first time, that the healing process in critical bone defects could be clearly improved by the implantation of these scaffolds considered as carriers of biological signals and bone-formers; in vivo assays suggested the potential of these Zn–MBG systems to enhance bone regeneration in the therapeutic practice.

The second approach was to combine the enrichment with antibacterial ions and the antibiotic loading in MBG scaffolds. In this sense, the study of Heras et al. [19] evaluated the antibacterial properties of gelatin-coated meso–macroporous scaffolds based on the 80%SiO_2_-15%CaO-5%P_2_O_5_ (mol.%) bioactive glass with a 4% of ZnO and loaded with either saturated or the minimal inhibitory concentrations of one of the following antibiotics: levofloxacin, vancomycin, rifampicin or gentamicin (Figure 9). Antibacterial in vitro studies with *E. coli* and *S. aureus* as bacteria models showed a synergistic effect of zinc ions and antibiotics. The effect was especially intense in planktonic cultures and biofilm total destruction of *E. coli* was accomplished with MBG scaffolds containing 4% of ZnO loaded with levofloxacin or gentamicin. In the case of *S. aureus,* in planktonic cultures inhibition, the best combination was scaffolds loaded with levofloxacin, or vancomycin. In addition, *S. aureus* biofilms were destroyed also under rifampicin.

This approach could be an important step in the fight against microbial resistance and provide needed options for bone infection treatment. All these studies showed that multifunctional 3D scaffolds could be fundamental tools in bone tissue engineering since they are able to remove bacterial infections with combination and lower antibiotic dosage and incorporation of therapeutic ions.

## 6. Mesoporous Silica Nanoparticles (MSNs) Scaffolds

Among the different types of inorganic nanomaterials, MSNs have been highlighted as promising multifunctional platforms for nanomedicine [167]. Since their introduction in the drug delivery scenery in 2001, nanoparticles have been proposed and studied extensively to control the release of different types of drugs. In this context, one of the strategies followed by Prof. Vallet-Regí was the inclusion of nanoparticles within scaffolds as a powerful that ensures a targeted tool in bone tissue engineering delivery, the biomolecule preservation and allows the different release kinetics that each of the included molecules required for their optimal beneficial effect (Figure 3) [168,169]. In this context and, as it was previously mentioned, MSNs have been proposed as drug carriers due to their high loading capacity, physicochemical stability and excellent biocompatibility. One of the main advantages of including these charged nanoparticles in scaffolds derives from local delivery to the tissue, which increases therapeutic efficacy by allowing dose reduction compared to systemic administration [98].

In addition, MSNs can be functionalized with molecular/supramolecular assemblies on their outer surface to develop closed MSNs that exhibit “zero-delivery” (i.e., the hybrid material is unable to release payload), but are able to release their payload in response to external stimuli [165,170]. The huge potential of designing a hybrid material with nanometer-controlled delivery characteristics has become an excellent strategy to develop MSN-capped systems to be opened by chemical or temperature changes, magnetic fields or biochemical triggers. The Prof. Vallet-Regí’s group in collaboration with Martínez-Mañez’s group developed “gated scaffolds” which consists of combining capped MSNs and classical porous biomaterials [171]. In particular, functionalized MSNs were prepared with amines and capped with ATP that could be selectively opened with an APase enzyme. Capped MSNs were integrated into gelatin scaffolds prepared by rapid prototyping (Figure 1). The obtained “gated scaffold” remained tightly capped in competitive aqueous buffered solutions is able to deliver the cargo in the presence of APase, an enzyme whose concentration is used to assess osteoclast activity in bone remodeling processes and as a marker for bone metastases. This study focused on the capability of active gelatin-based 3D macroporous scaffolds for on-command cargo delivery in the presence of APase.

On the other hand, Paris et al. [32] developed 3D porous scaffolds based on agarose and CHA, two structural components that act as a temporary mineralized extracellular matrix. The scaffolds were prepared by the GELPOR3D method (Figure 1). An angiogenic protein, VEGF, and an antibiotic, cephalexin, loaded in MSNs, were included to design multifunctional scaffolds for bone reconstruction. These scaffolds were suitable for the adhesion of preosteoblast cells, exhibiting a sustained cephalexin delivery adequate for inhibiting bacterial growth as well as a release of the proangiogenic molecule which induces blood vessel formation in chicken embryos grown ex ovo.

Using nanoparticles systems allows combination of gated or covered nanoparticles that can be opened at will using chemical, physical, or biochemical stimuli with different scaffold supports based on ceramics or polymers, opens up the possibility of preparing a number of advanced gated scaffolds, which help to find applications in regenerative medicine.

## 7. Conclusions and Future Perspectives

Prof. Vallet-Regí has been a pioneer in overcoming the barriers between disciplines: chemistry, pharmacy, biology, biotechnology, engineering, etc., contributing to the advancement of knowledge in the designing and manufacturing of scaffolds for bone tissue engineering. Although it has not been an easy task to reflect such a considerable and stimulating effort, the authors hope they have succeeded in describing the road travelled from the synthesis of calcium phosphate ceramics, glasses, mesoporous bioactive glasses and mesoporous particles to their integration with other organic components in the form of hierarchical 3D-connected porous scaffolds improved by the inclusion of different biomolecules and/or through their surface functionalization.

Throughout this review, the manufacturability of scaffolds for bone tissue engineering as well as the ability to regenerate new bone tissue when bioactive molecules were incorporated has been demonstrated. A fundamental challenge in the field of hard tissue engineering by 3D scaffolding is the gap that still exists between construct fabrication and clinical practice in order to offer actual solutions for a concrete pathology in a given patient. In this sense, researchers must have established well the differences between in vitro experiments and in vivo practice due to mainly tissue regeneration being a dynamic process that involves a bidirectional interaction between the biological environment and the surrounding matrix. Enriching this interaction and developing smart responses is an important challenge could be fulfilled by creating new bioinspired inks for bone engineering which mimic the chemistry and arrangement of apatite nanocrystals, native bone extracellular matrix compounds, as well as the diverse biochemical signals needed to stimulate bone growth. For this purpose, it is necessary to develop a technique that ensures a sufficient number of initial cells are used to create the critical volume to build bone tissues, as well as the inclusion of therapeutic agents, drugs, biological signals together with living cells without affecting their functionality. A major goal is to mimic this dynamic reciprocity fostered by biomaterial designs that adapt to local cellular changes and can alter their properties in response to local biological signals inducing new bone formation.

## Figures and Tables

**Figure 1 pharmaceutics-13-01981-f001:**
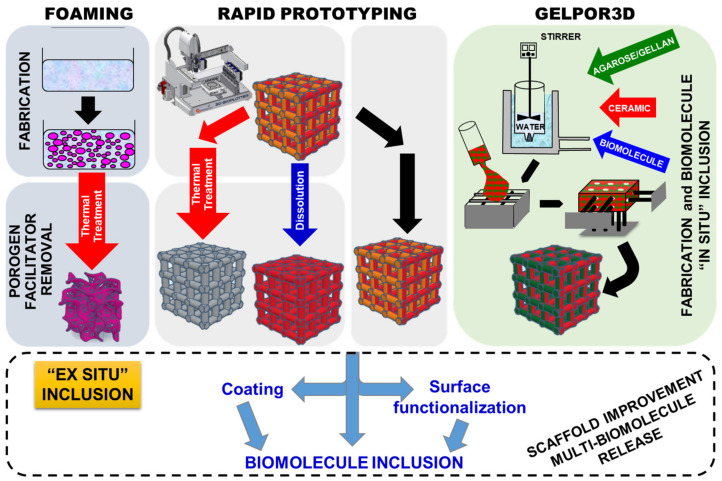
Schematic representation of the different scaffolds fabrication technologies employed as well as the additional routes designed to improve and tailor their final performance.

**Figure 2 pharmaceutics-13-01981-f002:**
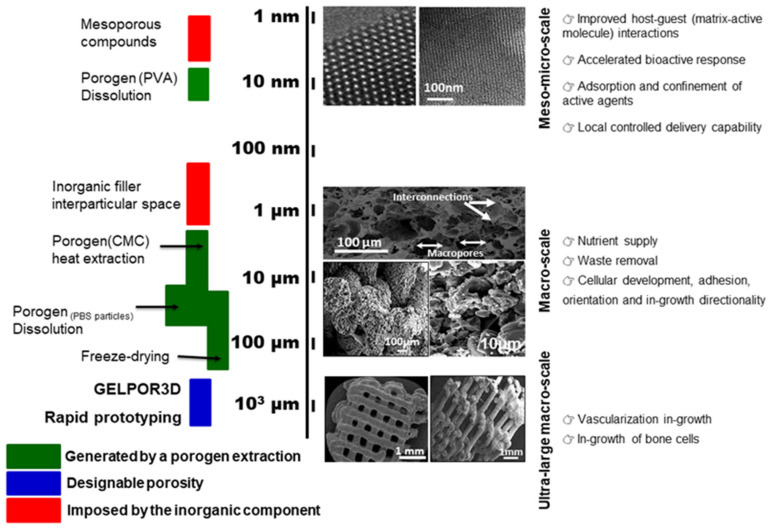
Schematic representation, related to the length scale, of the scaffolds pore distribution, the methods employed to obtain it, micrographs at different magnifications and the different physiological phenomena taking place (SEM micrographs reproduced with permission from Shruti et al. [10], Acta Biomater.; published by Elsevier, 2013 and Heras et al. [47], Acta Biomater.; published by Elsevier, 2019).

**Figure 3 pharmaceutics-13-01981-f003:**
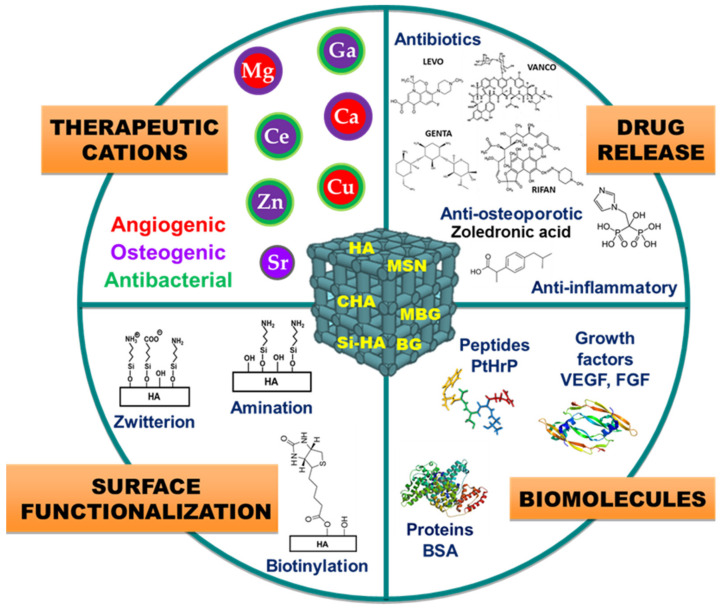
Representation of bioceramic scaffold modifications for bone regeneration.

**Figure 4 pharmaceutics-13-01981-f004:**
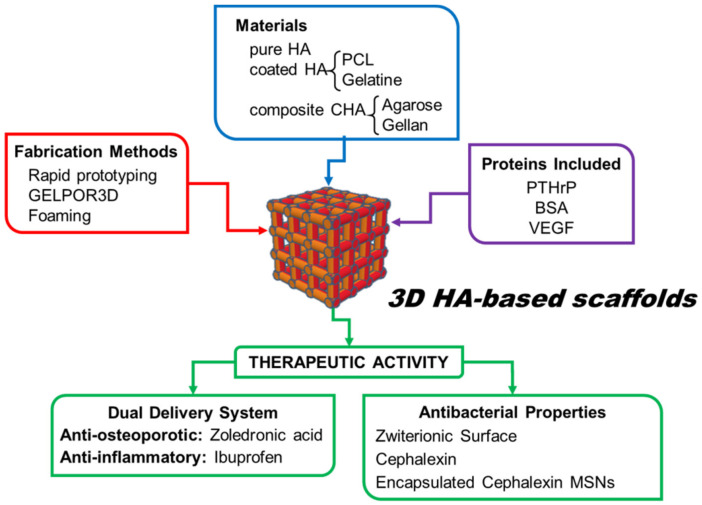
Strategies carried out in the fabrication of engineered HA-based scaffolds.

**Figure 5 pharmaceutics-13-01981-f005:**
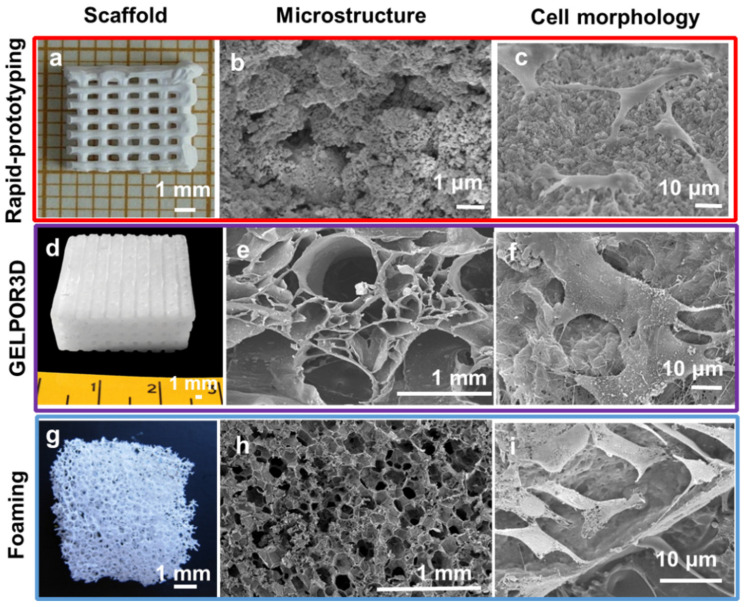
Photographs of HA-based scaffolds fabricated by three different methods (**a**,**d**,**g**); scanning electron micrographs showing the scaffold microstructure (**b**,**e**,**h**) and the morphology of osteoblast-like cells cultured onto the scaffolds (**c**,**f**,**i**). Photographs and SEM micrographs reproduced with permission from Sánchez-Salcedo et al. [60], J. Mater. Chem. B; published by Royal Society of Chemistry, 2013; Paris et al. [34], Int. J. Pharm.; published by Elsevier, 2015; Alcaide et al. [28], J. Biomed. Mater. Res. Part A; published by Wiley, 2010; Sánchez-Salcedo et al. [37], *J. Mater. Chem.*; published by Royal Society of Chemistry, 2010; Ardura et al. [39], J. Biomed. Mater. Res. Part A; published by Wiley, 2016; Cicuéndez et al. [36], Acta Biomater.; published by Elsevier, 2012; Adapted from [30], IOP Publishing, 2011.

**Figure 6 pharmaceutics-13-01981-f006:**
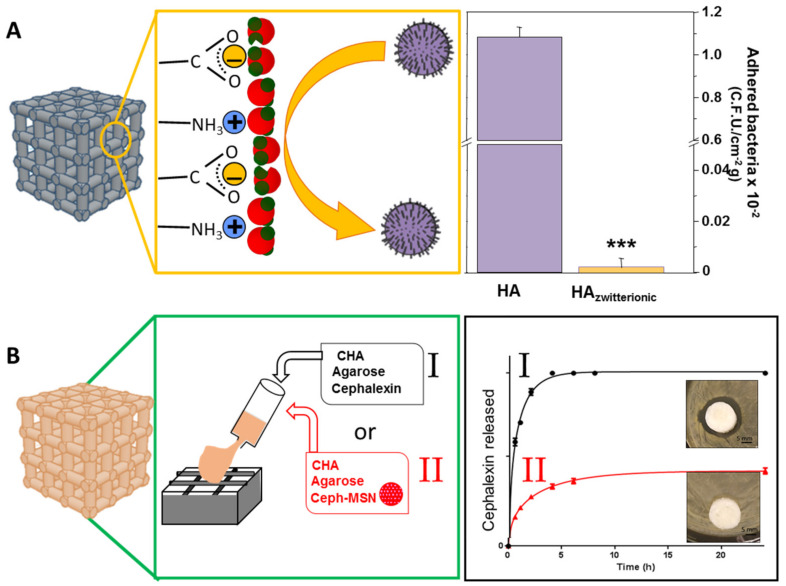
Strategies used to combat bone infection: (**A**) (left) schematic representation of the water molecules disposition on the zwitterionic surface that generates a strong repulsive force against bacteria as they approach the surface. (right) Bacteria attached onto HA and HA-Zwitterionic scaffold surfaces. (***) Statistical significance *p* < 0.001. Reproduced with permission from Sánchez-Salcedo et al. [60], J. Mater. Chem. B; published by Royal Society of Chemistry, 2013. (**B**) (left) Two methodologies (I and II) designed for the incorporation of antibiotics into the scaffold. (right) Release profiles from these scaffolds and agar diffusion tests. Reproduced with permission from Paris et al. [32], Acta Biomater.; published by Elsevier, 2019.

**Figure 7 pharmaceutics-13-01981-f007:**
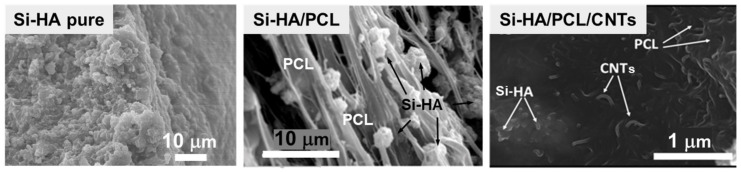
SEM micrographs at high magnifications inside filaments of Si-HA pure scaffold obtained through thermal treatment by removed MMA + MBAA (**left**), and two different Si-HA composites: Si-HA/PCL where Si-HA was embedded in PLC fibers ((**middle**), reproduced with permission from Meseguer-Olmo et al. [120], J. Biomed. Mater. Res. Part A; published by Wiley, 2013) and Si-HA/PCL/CNTs where CNTs and Si-HA were homogeneously dispersed in organic phase ((**right**), reproduced with permission from Gonçalves [121], J. Biomed. Mater. Res. Part B Appl. Biomater.; published by Wiley, 2015).

**Figure 8 pharmaceutics-13-01981-f008:**
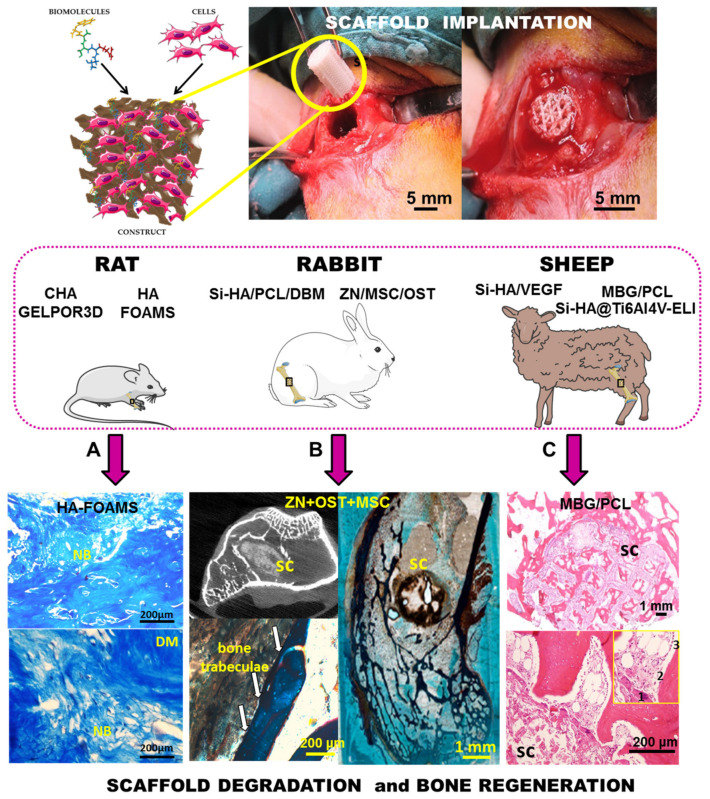
Illustration of the construct upgraded with biomolecules and cells. Photograph of its surgical implantation and the subsequent blood soaking within the recipient rabbit bone. Representative µCT projection images and histological studies of the areas surrounding the different scaffolds (sc) implanted in different animal models: (**A**) CHA-GELPORE3D scaffolds extracted from subcutaneous implants of rat after 21 or 30 days and stained (hematoxylin–eosin): (1) skin, (2) fibrous capsule, (3) blood vessel, reproduced with permission from Ardura et al. [39], J. Biomed. Mater. Res. Part A; published by Wiley, 2016. (**B**) Masson–Goldner Trichrome sagittal stained sections of the area surrounding ZN+OST+MSC scaffolds loaded with osteostatin and decorated with MSCs, showing newly formed bone after 3 months of defect generation in the rabbit femur. In detail, bone trabeculae formed in intimate contact with the degraded material suggesting osseointegration (arrows), reproduced with permission from Lozano et al. [20], J. Mater. Sci. Mater. Med.; published by Springer, 2020. (**C**) Histological examination of MBG/PCL after 12 weeks of implantation. The inset shows (1) osteoclast cells, (2) blood vessel and (3) the osteoblast border, reproduced with permission from Gómez-Cerezo et al. [17], Acta Biomater.; published by Elsevier, 2019.

**Figure 9 pharmaceutics-13-01981-f009:**
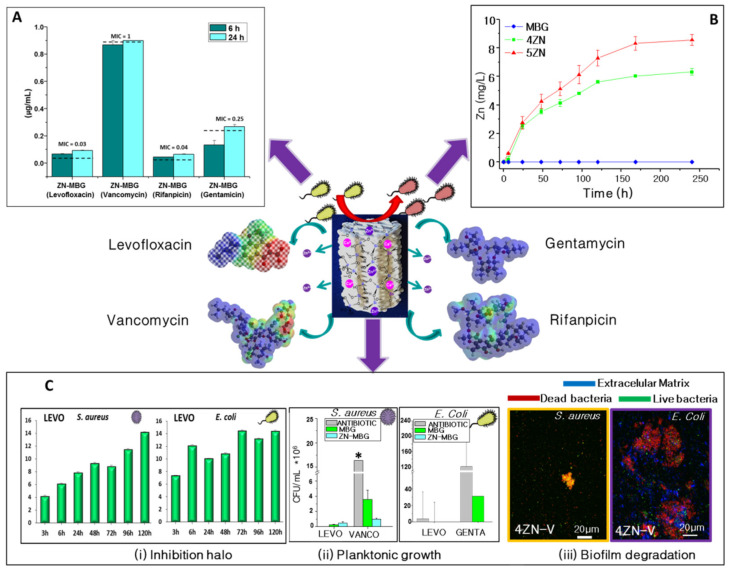
Representative scheme of Zn-MBG scaffolds loaded with antibiotics. (**A**) Antibiotic release from Zn-MBG scaffolds at physiological pH after 6 and 24 h (MIC, Minimum inhibitory concentration). (**B**) Evolution of zinc release from 4% ZnO (4ZN-MBG) and 5% ZnO (5ZN-MBG) enriched MBG scaffolds in the THB medium. (**C**) (i) Levofloxacin-loaded scaffold inhibition rate against *S. aureus* and *E. coli* at pH 7.4 (physiological environment). (ii) Planktonic growth of *S. aureus* and *E. Coli* in contact with antibiotic (Levofloxacin, vancomycin and gentamicin)-loaded MBG and Zn-MBG scaffolds * indicate significant differences between samples where the antibiotic is loaded within the scaffold and those where the drug is dissolved in the medium. Statistical significance: *p* < 0.05. (iii) Biofilm degradation with vancomycin-loaded Zn-MBGscaffolds after 24h. Reproduced with permission from Heras et al. [19], Acta Biomater.; published by Elsevier, 2020.

**Table 1 pharmaceutics-13-01981-t001:** 3D scaffolds formed by the ceramic matrix of Si-HA and manufactured by the rapid prototyping method.

Scaffold Type	Organic Polymer (Binder Agent) *	Other Subsequent Incorporations	Effects	Ref.
Si-HA	Hydroxypropyl methylcellulose **		Higher surface areas and porosities that facilitate protein adsorption (albumin and fibrinogen)	[117]
	MMA+MBAA **	Surface functionalization with biotin by covalent bonding	Possibility of further anchoring of more molecules from the biotin present on the scaffold surfaces that could act as a linker	[7]
		Osteostatin incorporation by adsorption or covalently anchored	Regardless of the osteostatin incorporation strategy, its presence stimulates preosteoblast cell growth and matrix mineralization	[115]
		Fibroblast growth factors (FGFs) adsorption	Improve adhesion and proliferation of osteoblastic cells.	[9]
		Elastin-like recombinamers (ELRs) adsorption.	Induce bone marrow mesenchymal stromal cells proliferation and differentiation into osteoblastic lineage	[116]
NanoSi-HA	PVA ***	Vascular endothelial growth factor (VEGF) adsorption.	Poor results but VEGF presence increased volume of newly formed bone, trabeculae thickness and implant vascularization in sheep in vivo model	[118]
Si-HA/Gelatine	Gelatine type A	Vancomycin	Improve pre-osteoblatic cells differentiation and ALP gene expression. Cargo-antibiotic agent for drug delivery.	[119]
Si-HA/PCL	PCL		Good results of biocompatibility, osteroconductive features and new bone formation capability in in vivo New Zealand rabbit’s studies	[120]
Si-HA/PCL/DBM		Incorporation of demineralized bone matrix (DMB)	In vivo New Zealand rabbit’s studies: new bone formation in the peripheral portions of the scaffolds and within its pores.	
Si-HA/PCL/CNTs		Incorporation of carbon nanotubes (CNTs)	Improve protein adsorption and osteoblast-like cells attachment.	[121]
Si-HA/PVA	PVA	Vascular endothelial growth factor (VEGF) adsorption.	Stimulated endothelial (EC_2_) proliferation and pre-osteoblasts (MC3T3-E1) differentiation	[118]

(*) Organic polymer (binder agent) removed by thermal treatment at (**) 600–700 °C or (***) 1150 °C.

**Table 2 pharmaceutics-13-01981-t002:** The 3D scaffolds formed by ceramic matrix of MBGs and manufactured by rapid prototyping method.

Scaffold Type	Organic Polymer (Binder Agent)	Scaffold Modification	Effects	Ref.
^(a)^ MGHA	Hydroxy methylcellulose	Nano HA embedded and amine functionalization	Enhanced preosteoblast adhesion, proliferation and differentiation	[11]
MBG/PCL	PCL	PBS particles	Extra microporous Increase bioactivity and neovascularization	[22]
		Zoledronic acid loaded	Antiresorptive and avoids inflammatory response	[17]
^(b)^ 4Zn-MBG *	PCL/Gelatine crosslinked GA	Osteostatin	Osteogenic	[20,47]
		Osteostatin and MSCs	Significantly improved trabecular bone volume density from μCT	[20]
^(a)^ MGHA	Hydroxy methylcelullose	Antibiotic loading (Levofloxacin)	pH-dependent Levofloxacin release is able to inhibit the *S. aureus* growth and to destroy a preformed biofilm	[16]
G_RIF_MG_LEV_PVA_VAN_	PVA	Antibiotic loading (Levofloxacin, Rifanpicin, Vancomicin)	Multidrug scaffolds release	[15]
^(b)^ X-MBG * X = Ce, Ga, Zn, Sr	PCL and/or Gelatine crosslinked GA	Therapeutic ions	Osteogenic, angiogenic and antimicrobial	[10,19,21]
^(b)^ 4Zn-MBG *	PCL/Gelatine crosslinked GA	Antibiotic loading (Levofloxacin, Rifanpicin, Vancomicin and Gentamicin)	Eliminates *Staphylococcus* and *Escherichia* biofilms and inhibits bacteria growth in very short time periods	[19]

^(a)^ Organic polymer (binder agent) removed by thermal treatment at 700 °C or ^(b)^ polymer removal and posterior gelatine-crosslinked GA covering. * 4% ZnO enriched MBG.

## Data Availability

Not applicable.

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
