# Peer review of "Design of 3D Scaffolds for Hard Tissue Engineering: From Apatites to Silicon Mesoporous Materials"

_pharmaceutics, 2021, doi:10.3390/pharmaceutics13111981_

Round 1
Reviewer 1 Report
The design of 3D scaffolds for hard tissue engineering is a much wider field than the representation in manuscript. The rapid prototyping technique is a much earlier known technique:
- Rüdiger Landers, Ute Hübner, Rainer Schmelzeisen, Rolf Mülhaupt, Rapid prototyping of scaffolds derived from thermoreversible hydrogels and tailored for applications in tissue engineering, Biomaterials, Volume 23, Issue 23, 2002, Pages 4437-4447, ISSN 0142-9612, https://doi.org/10.1016/S0142-9612(02)00139-4.
- H Ang, F.S.A Sultana, D.W Hutmacher, Y.S Wong, J.Y.H Fuh, X.M Mo, H.T Loh, E Burdet, S.H Teoh, Fabrication of 3D chitosan–hydroxyapatite scaffolds using a robotic dispensing system, Materials Science and Engineering: C, Volume 20, Issues 1–2, 2002, Pages 35-42, ISSN 0928-4931, https://doi.org/10.1016/S0928-4931(02)00010-3.
The manuscripts that form the basis of this technique are not cited in the submitted manuscript.
The manuscript shows the results of only one part of this technique (by one research group).
A broader overview of this issue is already known in the literature:
- Qu, M., Wang, C., Zhou, X., Libanori, A., Jiang, X., Xu, W., Zhu, S., Chen, Q., Sun, W., Khademhosseini, A., Multi-Dimensional Printing for Bone Tissue Engineering. Healthcare Mater.2021, 10, 2001986. https://doi.org/10.1002/adhm.202001986
- Susheem Kanwar, Sanjairaj Vijayavenkataraman,Design of 3D printed scaffolds for bone tissue engineering: A review, Bioprinting, Volume 24, 2021, e00167, ISSN 2405-8866, https://doi.org/10.1016/j.bprint.2021.e00167.
- Niyou Wang, S. Thameem Dheen, Jerry Ying Hsi Fuh, A. Senthil Kumar, A review of multi-functional ceramic nanoparticles in 3D printed bone tissue engineering, Bioprinting, Volume 23, 2021, e00146, ISSN 2405-8866, https://doi.org/10.1016/j.bprint.2021.e00146
- Chong Wang, Wei Huang, Yu Zhou, Libing He, Zhi He, Ziling Chen, Xiao He, Shuo Tian, Jiaming Liao, Bingheng Lu, Yen Wei, Min Wang, 3D printing of bone tissue engineering scaffolds, Bioactive Materials, Volume 5, Issue 1, 2020, Pages 82-91, ISSN 2452-199X, https://doi.org/10.1016/j.bioactmat.2020.01.004.
- Maria P. Nikolova, Murthy S. Chavali,Recent advances in biomaterials for 3D scaffolds: A review, Bioactive Materials, Volume 4, 2019,Pages 271-292, ISSN 2452-199X,https://doi.org/10.1016/j.bioactmat.2019.10.005.
- Lei Zhang, Guojing Yang, Blake N. Johnson, Xiaofeng Jia, Three-dimensional (3D) printed scaffold and material selection for bone repair, Acta Biomaterialia, Volume 84, 2019, Pages 16-33, ISSN 1742-7061, https://doi.org/10.1016/j.actbio.2018.11.039
The manuscript is a kind of hymn to one author (which is undoubtedly of high quality). This should be avoided in the text of the manuscript (quote the author in the references).
Author Response
REFEREE 1
The authors really appreciate the reviewer’s comments.
The design of 3D scaffolds for hard tissue engineering is a much wider field than the representation in manuscript. The rapid prototyping technique is a much earlier known technique:
- Rüdiger Landers, Ute Hübner, Rainer Schmelzeisen, Rolf Mülhaupt, Rapid prototyping of scaffolds derived from thermoreversible hydrogels and tailored for applications in tissue engineering, Biomaterials, Volume 23, Issue 23, 2002, Pages 4437-4447, ISSN 0142-9612, https://doi.org/10.1016/S0142-9612(02)00139-4.
- H Ang, F.S.A Sultana, D.W Hutmacher, Y.S Wong, J.Y.H Fuh, X.M Mo, H.T Loh, E Burdet, S.H Teoh, Fabrication of 3D chitosan–hydroxyapatite scaffolds using a robotic dispensing system, Materials Science and Engineering: C, Volume 20, Issues 1–2, 2002, Pages 35-42, ISSN 0928-4931, https://doi.org/10.1016/S0928-4931(02)00010-3.
Thanks for this information, these articles have been included in the revised manuscript. References: 43 and 44.
The manuscripts that form the basis of this technique are not cited in the submitted manuscript.
We really appreciate the referee’s comments. The authors have included additional works regarding rapid prototyping technique. These manuscripts have been included in the text as the referee suggests. References: 41, 42, 45, 46 167.
The manuscript shows the results of only one part of this technique (by one research group).
Due to review is a commemorative Issue in honour of Professor María Vallet Regí "20 Years of Silica-Based Mesoporous Materials" the authors have focused on the contribution of her research group to this technique. However, following the recommendations of the reviewer we have included the following articles (marked in red in the new version):
A broader overview of this issue is already known in the literature:
- Qu, M., Wang, C., Zhou, X., Libanori, A., Jiang, X., Xu, W., Zhu, S., Chen, Q., Sun, W., Khademhosseini, A., Multi-Dimensional Printing for Bone Tissue Engineering. Healthcare Mater. 2021, 10, 2001986. https://doi.org/10.1002/adhm.202001986
- Susheem Kanwar, Sanjairaj Vijayavenkataraman,Design of 3D printed scaffolds for bone tissue engineering: A review, Bioprinting, Volume 24, 2021, e00167, ISSN 2405-8866, https://doi.org/10.1016/j.bprint.2021.e00167.
- Niyou Wang, S. Thameem Dheen, Jerry Ying Hsi Fuh, A. Senthil Kumar, A review of multi-functional ceramic nanoparticles in 3D printed bone tissue engineering, Bioprinting, Volume 23, 2021, e00146, ISSN 2405-8866, https://doi.org/10.1016/j.bprint.2021.e00146
- Chong Wang, Wei Huang, Yu Zhou, Libing He, Zhi He, Ziling Chen, Xiao He, Shuo Tian, Jiaming Liao, Bingheng Lu, Yen Wei, Min Wang, 3D printing of bone tissue engineering scaffolds, Bioactive Materials, Volume 5, Issue 1, 2020, Pages 82-91, ISSN 2452-199X, https://doi.org/10.1016/j.bioactmat.2020.01.004.
- Maria P. Nikolova, Murthy S. Chavali,Recent advances in biomaterials for 3D scaffolds: A review, Bioactive Materials, Volume 4, 2019,Pages 271-292, ISSN 2452-199X,https://doi.org/10.1016/j.bioactmat.2019.10.005.
- Lei Zhang, Guojing Yang, Blake N. Johnson, Xiaofeng Jia, Three-dimensional (3D) printed scaffold and material selection for bone repair, Acta Biomaterialia, Volume 84, 2019, Pages 16-33, ISSN 1742-7061, https://doi.org/10.1016/j.actbio.2018.11.039
The manuscript is a kind of hymn to one author (which is undoubtedly of high quality). This should be avoided in the text of the manuscript (quote the author in the references).
As we answer previously this review article is a commemorative Issue in honour of Professor MarÍa Vallet Regí “20 Years of Silica-Based Mesoporous Materials” and on page 2, the authors wrote “This article tries to compile the considerable effort carried out by Prof. Vallet-Regí’s group…..”.
For the avoidance of doubt as to the purpose of this review, we have included the following paragraph at the end of the Introduction and a new Figure 3: “This paper is a review article in honor of the scientific career of Prof. Vallet-Regí on scaffolding for bone regeneration area in the last 20 years. This review has tried to condense the part of her prolific scientific trajectory devoted to the design of tridimensional scaffolds that could efficiently solve actual pathologies arisen by clinicians. This manuscript has been organized as a function of inorganic matrices, calcium phosphates, glasses and mesoporous glasses, that constitutes the main component of the designed, fabricated, in vitro evaluated and in vivo implanted scaffolds. Figure 3 summarizes the different ceramic scaffolds described in this review and compiles the different modifications and/or incorporations made in these ceramic matrices”.

Reviewer 2 Report
This review paper titled “Design of 3D scaffolds for hard tissue engineering: From apatites to silicon mesoporous materials” (Pharmaceutics-1447117) was well accepted, I suggest the acceptance on the condition that some minor issues were improved.
- In the introduction section, such as page 2 and page 3, some references should be cited at proper place, please check the whole manuscript for this issue, and check the figure captions, some references should be cited as well.
- Page 10, line 378, there were some metal-substituted HA for antibacterial applications, please re-check the describe them in this section.
- Please cite recent related papers in this paper, such as https://www.mdpi.com/2073-4352/11/4/353.
Author Response
REFEREE 2
The authors really appreciate the reviewer’s comments.
This review paper titled “Design of 3D scaffolds for hard tissue engineering: From apatites to silicon mesoporous materials” (Pharmaceutics-1447117) was well accepted, I suggest the acceptance on the condition that some minor issues were improved.
- In the introduction section, such as page 2 and page 3, some references should be cited at proper place, please check the whole manuscript for this issue, and check the figure captions, some references should be cited as well.
Following the referee recommendations, the authors have checked the references which are marked in red in the revised manuscript.
- Page 10, line 378, there were some metal-substituted HA for antibacterial applications, please re-check the describe them in this section.
We agree with the referee. Although there are other strategies based in HA scaffolds for the infection treatment we describe just the approaches used by the Prof. Vallet-Regí. The manuscript has been modified to include other approaches with some references (page 10): “For this purpose, different strategies have been designed by using HA based scaffolds, such as, the doping with metal ions with antibacterial properties (e.g. Ag+, Cu2+, Zn2+, …) [90-93] or the design of hybrid scaffolds with intrinsic antimicrobial activity (e.g. chitosan-based scaffolds [94-96]). Alternatively, this challenge has also been addressed by Prof. Vallet-Regí’s group by using two different approaches which allow to combat bone infection and simultaneously to regenerate bone tissue (Figure 6)”
- Please cite recent related papers in this paper, such as https://www.mdpi.com/2073-4352/11/4/353.
This manuscript and other recent related papers have been included in the revised manuscript (for example references 45, 46, 65, 66,…).

Reviewer 3 Report
The review by Ana Garcia et al is important summary of work of prof. Regi. It is giving comprehensive information about materials for 3D scaffold meant for hard tissues.
There are several issues that could be improved:
Enhancing keywords for apatite modifying agents .
Page 3-4 has no references, there should be mentioned source.
Conclusion should mention the most significant outcomes , and give a future overview in the field.
Generally I would have expect more schema and table summarizing the work, and making it more pictorial.
Author Response
REFEREE 3
The authors really appreciate the reviewer’s comments.
The review by Ana Garcia et al is important summary of work of Prof. Vallet-Regí. It is giving comprehensive information about materials for 3D scaffold meant for hard tissues.
There are several issues that could be improved:
Enhancing keywords for apatite modifying agents.
Following the referee’s suggestion, we have incorporated the term “ceramic modifying agents” in the keywords section.
Page 3-4 has no references, there should be mentioned source.
We really appreciate the referee’s comments. Authors have included more references in the introduction section and we are marked in red in the new version.
Conclusion should mention the most significant outcomes, and give a future overview in the field.
Following the referee’s suggestion, we have incorporated a future overview. In the summary section the authors have changed by Conclusions and Future Perspectives and wehave inclided this paragraph:
“Throughout this review, the manufacturability of scaffolds for bone tissue engineering as well as the ability to regenerate new bone tissue when bioactive molecules were incor-porated has been demonstrated. A fundamental challenge in the field of hard tissue en-gineering by 3D scaffolding is the gap that still exists between construct fabrication and clinical practice in order to offer actual solutions for a concrete pathology in a given pa-tient. In this sense, researchers must have well established the differences between in vitro experiments and in vivo practice due to mainly tissue regeneration is a dynamic process that involve a bidirectional interaction between biological environment and the surrounding matrix. Enriching this interaction and developing smart responses is an important challenge could be fulfilling by creating new bioinspired inks for bone engi-neering which mimic the chemistry and arrangement of apatite nanocrystals, native bone extracellular matrix compounds, as well as the diverse biochemical signals needed to stimulate bone growth. For this purpose, it is necessary to develop a technique that en-sures a sufficient number of initial cells to create the critic volume to build bone tissues, as well as the inclusion of therapeutic agents, drugs, biological signals together with living cells without affecting their functionality. A major goal is to mimic this dynamic reciprocity fostered by biomaterial designs that adapt to local cellular changes and can alter their properties in response to local biological signals inducing new bone formation.”
Generally, I would have expect more schema and table summarizing the work, and making it more pictorial.
Following the referee’s suggestion, we have included four new figures (Figure 3, 6, 7 and 9) in the text and a new graphical abstract.
